# Mode Estimation for High Dimensional Discrete Tree Graphical Models

**Chao Chen**
Department of Computer Science
Rutgers, The State University of New Jersey
Piscataway, NJ 08854-8019
chao.chen.cchen@gmail.com

**Han Liu**
Department of Operations Research
and Financial Engineering
Princeton University, Princeton, NJ 08544
hanliu@princeton.edu

**Dimitris N. Metaxas**
Department of Computer Science
Rutgers, The State University of New Jersey
Piscataway, NJ 08854-8019
dnm@cs.rutgers.edu

**Tianqi Zhao**
Department of Operations Research
and Financial Engineering
Princeton University, Princeton, NJ 08544
tianqi@princeton.edu

## Abstract

This paper studies the following problem: given samples from a high dimensional discrete distribution, we want to estimate the leading $(\delta, \rho)$-modes of the underlying distributions. A point is defined to be a $(\delta, \rho)$-mode if it is a local optimum of the density within a $\delta$-neighborhood under metric $\rho$. As we increase the "scale" parameter $\delta$, the neighborhood size increases and the total number of modes monotonically decreases. The sequence of the $(\delta, \rho)$-modes reveal intrinsic topographical information of the underlying distributions. Though the mode finding problem is generally intractable in high dimensions, this paper unveils that, if the distribution can be approximated well by a tree graphical model, mode characterization is significantly easier. An efficient algorithm with provable theoretical guarantees is proposed and is applied to applications like data analysis and multiple predictions.

## 1 Introduction

Big Data challenge modern data analysis in terms of large dimension, insufficient sample and the inhomogeneity. To handle these challenges, new methods for visualizing and exploring complex datasets are crucially needed. In this paper, we develop a new method for computing *diverse modes* of the unknown discrete distribution function. Our method is applicable in many fields, such as computational biology, computer vision, etc. More specifically, our method aims to find a sequence of $(\delta, \rho)$-modes, which are defined as follows:

**Definition 1** ($(\delta, \rho)$-modes). *A point is a $(\delta, \rho)$-mode if and only if its probability is higher than all points within distance $\delta$ under a distance metric $\rho$.*

With a metric $\rho(\cdot)$ given, the $\delta$-*neighborhood* of a point $x$, $\mathcal{N}_\delta(x)$, is defined as the ball centered at $x$ with radius $\delta$. Varying $\delta$ from small to large, we can examine the topology of the underlying distribution at different scales. Therefore $\delta$ is also called the *scale* parameter. When $\delta = 0, \mathcal{N}_\delta(x) = \{x\}$, so every point is a mode. When $\delta = \infty$, $\mathcal{N}_\delta(x)$ is the whole domain, denoted by $\mathcal{X}$, so the maximum a posteriori is the only mode. As $\delta$ increases from zero to infinity, the $\delta$-neighborhood of $x$ monotonically grows and the set of modes, denoted by $\mathcal{M}^\delta$, monotonically decreases. Therefore as $\delta$ increases, the sets of $\mathcal{M}^\delta$ form a nested sequence, which can be viewed as a multi-scale description of the underlying probability landscape. See Figure 1 for an illustrative example. In this paper, we will use the *Hamming distance*, $\rho_{\mathrm{H}}$, i.e., the number of variables at which two points disagree. Other distance metrics, e.g., the $L_2$ distance $\rho_{\mathrm{L2}}(x, x') = \|x - x'\|_2$, are also possible but with more computational challenges.

The concept of modes can be justified by many practical problems. We mention the following two motivating applications: (1) Data analysis: modes of multiple scales provide a comprehensive

geometric description of the topography of the underlying distribution. In the low-dimensional continuous domain, such tools have been proposed and used for statistical data analysis [20, 17, 3]. One of our goals is to carry these tools to the discrete and high dimensional setting. (2) Multiple predictions: in applications such as computational biology [9] and computer vision [2, 6], instead of one, a model generates multiple predictions. These predictions are expected to have not only high probability but also high diversity. These solutions are valid hypotheses which could be useful in other modules down the pipeline. In this paper we address the computation of modes, formally,

**Problem 1** ($M$-modes). *For all $\delta$'s, compute the $M$ modes with the highest probabilities in $\mathcal{M}^\delta$.*

This problem is challenging. In the continuous setting, one often starts from random positions, estimates the gradient of the distribution and walks along it towards the nearby mode [8]. However, this gradient-ascent approach is limited to low-dimensional distributions over continuous domains. In discrete domains, gradients are not defined. Moreover, a naive exhaustive search is computationally infeasible as the total number of points is exponential to dimension. In fact, even deciding whether a given point is a mode is expensive as the neighborhood has exponential size.

In this paper, we propose a new approach to compute these discrete $(\delta, \rho)$-modes. We show that the problem becomes computationally tractable when we restrict to distributions with tree factor structures. We explore the structure of the tree graphs and devise a new algorithm to compute the top $M$ modes of a tree-structured graphical model. Inspired by the observation that a global mode is also a mode within smaller subgraphs, we show that all global modes can be discovered by examining all local modes and their consistent combinations. Our algorithm first computes local modes, and then computes the high probability combinations of these local modes using a junction tree approach. We emphasize that the algorithm itself can be used in many graphical model based methods, such as conditional random field [10], structured SVM [22], etc.

When the distribution is not expressed as a factor graph, we will first estimate the tree-structured factor graph using the algorithm of Liu *et al*. [13]. Experimental results demonstrate the accuracy and efficiency of our algorithm. More theoretical guarantee of our algorithm can be found in [7].

**Related work.** Modes of distributions have been studied in continuous settings. Silverman [21] devised a test of the null hypothesis of whether a kernel density estimation has a certain number of modes or less. Modes can be used in clustering [8, 11]. For each data point, a monotonically increasing path is computed using a gradient-ascend method. All data points whose gradient path converge to a same mode is labeled the same class. Modes can be also used to help decide the number of mixture components in a mixture model, for example as the initialization of the maximum likelihood estimation [11, 15]. The topographical landscape of distributions has been studied and used in characterizing topological properties of the data [4, 20, 17]. Most of these approaches assume a kernel density estimation model. Modes are detected by approximating the gradient using $k$-nearest neighbors. This approach is known to be inaccurate for high dimensional data.

We emphasize that the multi-scale view of a function has been used broadly in compute vision. By convolving an image with a Gaussian kernel of different widths, we obtain different level of details. This theory, called the *scale-space theory* [25, 12], is used as the fundamental principle of most state-of-the-art image feature extraction techniques [14, 16]. This multi-scale view has been used in statistical data analysis by Chaudhuri and Marron [3]. Chen and Edelsbrunner [5] quantitatively measured the topographical landscape of an image at different scales.

Chen *et al*. [6] proposed a method to compute modes of a simple chain model. However, restricting to a simple chain will limit our mode prediction accuracy. A simple chain model has much less flexibility than tree-factored models. Even if the distribution has a chain structure, recovering the chain from data is computationally intractable: the problem requires finding the chain with maximal total mutual information, and thus is equivalent to the NP-hard travelling salesman problem.

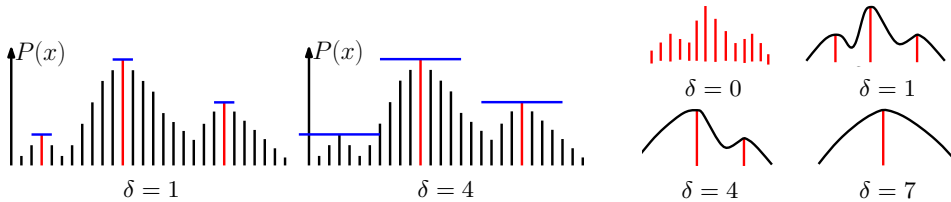

Figure 1: An illustration of modes of different scales. Each vertical bar corresponds to an element. The height corresponds to its probability. Left: when $\delta = 1$, there are three modes (red). Middle: when $\delta = 4$, only two modes left. Right: the multi-scale view of the landscape.

## 2  Background

**Graphical models.** We briefly introduce graphical models. Please refer to [23, 19] for more details. The graphical model is a powerful tool to model the joint distribution of a set of interdependent random variables. The distribution is encrypted in a graph $G = (\mathcal{V}, \mathcal{E})$ and a potential function $f$. The set of vertices/nodes $\mathcal{V}$ corresponds to the set of discrete variables $i \in [1, D]$, where $D = |\mathcal{V}|$. A node $i$ can be assigned a label $x_i \in \mathcal{L}$. A label configuration of all variables $x = (x_1, \ldots, x_D)$ is called a *labeling*. We denote by $\mathcal{X} = \mathcal{L}^D$ the domain of all labelings. The potential function $f : \mathcal{X} \to \mathbb{R}$ assigns to each labeling a real value, which is inversely proportional to the logarithm of the probability distribution, $p(x) = \exp(-f(x) - A)$, where $A = \log \sum_{x \in \mathcal{X}} \exp(-f(x))$ is the log-partition function. Thus the maximal modes of the distribution and the minimal modes of $f$ have a one-to-one correspondence. Assuming these variables satisfy the Markov properties, the potential function can be written as

$$f(x) = \sum_{(i,j) \in \mathcal{E}} f_{i,j}(x_i, x_j), \tag{2.1}$$

where $f_{i,j} : \mathcal{L} \times \mathcal{L} \to \mathbb{R}$ is the potential function for edge $(i, j)$ [1]. For convenience, we assume any two different labelings have different potential function values.

We define the following notations for convenience. A vertex subset, $\mathcal{V}' \subseteq \mathcal{V}$, *induces* a subgraph consisting of $\mathcal{V}'$ together with all edges whose both ends are within $\mathcal{V}'$. In this paper, all subgraphs are vertex-induced. Therefore, we abuse the notation and denote both the subgraph and the vertex subset by the same symbol.

We call a labeling of a subgraph $B$ a *partial labeling*. For a given labeling $y$, we may denote by $y_B$ its label configurations of vertices of $B$. We say the distance between two partial labelings $x_B$ and $y_{B'}$ is equal to the Hamming distance between the two within the intersection of the two subgraphs $\hat{B} = B \cap B'$, formally, $\rho(x_B, y_{B'}) = \rho(x_{\hat{B}}, y_{\hat{B}})$. We denote by $f_B(y_B)$ the potential of the partial labeling, which is only evaluated over edges within $B$. When the context is clear, we drop the subscript $B$ and write $f(y_B)$.

**Tree density estimation.** In this paper, we focus on tree-structured graphical models. A distribution that is Markov to a tree structure has the following factorization:

$$P(X = x) = p(x) = \prod_{(i,j) \in \mathcal{E}} \frac{p(x_i, x_j)}{p(x_i)p(x_j)} \prod_{k \in \mathcal{V}} p(x_k). \tag{2.2}$$

It is easy to see that the potential function can be written in the form (2.1). In the case when the input is a set of samples, we will first use the tree density estimation algorithm [13] to estimate the graphical model. The *oracle tree distribution* is the one on the space of all tree distributions that minimizes the Kullback-Leibler (KL) divergence between itself and the tree density, that is, $q^* = \operatorname{argmin}_{q \in \mathcal{P}_T} D(p^* || q)$, where $\mathcal{P}_T$ is the family of distributions supported on a tree graph, $p^*$ is the true density, and $D(p || q) = \sum_{x \in \mathcal{X}} p(x)(\log p(x) - \log q(x))$ is the KL divergence. It is proved [1] that $q^*$ has the same marginal univariate and bivariate distribution as $p^*$. Hence to recover $q^*$, we only need to recover the structure of the tree. Denote by $\mathcal{E}^*$ the edge set of the oracle tree. Simple calculation shows that $D(p^* || q^*) = -\sum_{(i,j) \in \mathcal{E}^*} I_{ij} + \text{const}$, where

$$I_{ij} = \sum_{x_i=1}^{L} \sum_{x_j=1}^{L} p^*(x_i, x_j)(\log p^*(x_i, x_j) - \log p^*(x_i) - \log p^*(x_j)) \tag{2.3}$$

is called the mutual information between node $i$ and $j$. Therefore we can apply Kruskal's maximum spanning tree algorithm to obtain $\mathcal{E}^*$, with edge weights being the mutual information.

In reality, we do not know the true marginal univariate and bivariate distribution. We thus compute estimators $\hat{I}_{ij}$ from the data set $\{X^{(1)}, \ldots, X^{(n)}\}$ by replacing $p^*(x_i, x_j)$ and $p^*(x_i)$ in (2.3) with their estimates $\hat{p}(x_i, x_j) = \frac{1}{n} \sum_{s=1}^{n} \mathbb{1}\{X_i^{(s)} = x_i, X_j^{(s)} = x_j\}$ and $\hat{p}(x_i) = \frac{1}{n} \sum_{s=1}^{n} \mathbb{1}\{X_i^{(s)} = x_i\}$. The tree estimator is thus obtained by Kruskal's algorithm:

$$\hat{T}_n = \operatorname{argmax}_T \sum_{(i,j) \in \mathcal{E}(T)} \hat{I}_{ij}. \tag{2.4}$$

By definition, the potential function on each edge can be estimated similarly using the estimated marginal univariate and bivariate distributions. By (2.1), we have $\hat{f}(x) = \sum_{(i,j) \in \mathcal{E}(\hat{T})} \hat{f}_{i,j}(x_i, x_j)$, where $\hat{T}$ is the estimated tree using Kruskal's algorithm.

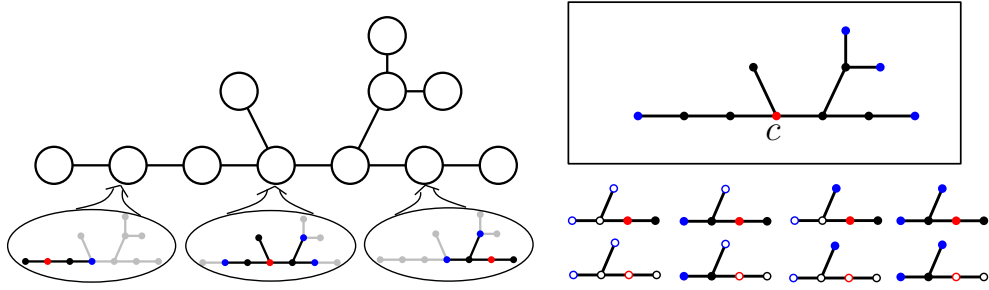

Figure 2: Left: The junction tree with radius $r = 2$. We show the geodesic balls of three supernodes. In each geodesic ball, the center is red. The boundary vertices are blue. The interior vertices are black and red. Right-bottom: Candidates of a geodesic ball. Each column corresponds to candidates of one boundary labeling. Solid and empty vertices represent label zero and one. Right-top: A geodesic ball with radius $r = 3$.

## 3 Method

We present the first algorithm to compute $\mathcal{M}^\delta$ for a tree-structured graph. To compute modes of all scales, we go through $\delta$'s from small to large. The iteration stops at a $\delta$ with only a single mode.

We first present a polynomial algorithm for the verification problem: deciding whether a given labeling is a mode (Sec. 3.1). However, this algorithm is insufficient for computing the top $M$ modes because the space of labelings is exponential size. To compute global modes, we decompose the problem into computing modes of smaller subgraphs, which are called *local modes*. Because of the bounded subgraph size, local modes can be solved efficiently. In Sec. 3.2, we study the relationship between global and local modes. In Sec. 3.3 and Sec. 3.4, we give two different methods to compute local modes, depending on different situations.

### 3.1 Verifying whether a labeling is a mode

To verify whether a given labeling $y$ is a mode, we check whether there is another labeling within $\mathcal{N}_\delta(y)$ with a smaller potential. We compute the labeling within the neighborhood with the minimal potential, $y^* = \mathrm{argmin}_{z \in \mathcal{N}_\delta(y)} f(z)$. The given labeling $y$ is a mode if and only if $y^* = y$.

We present a message-passing algorithm. We select an arbitrary node as the root, and thus a corresponding child-parent relationship between any two adjacent nodes. We compute messages from leaves to the root. Denote by $T_j$ as the subtree rooted at node $j$. The message from vertex $i$ to $j$, $\mathrm{MSG}_{i \to j}(\ell_i, \tau)$ is the minimal potential one can achieve within the subtree $T_i$ given a fixed label $\ell_i$ at $i$ and a constraint that the partial labeling of the subtree is no more than $\tau$ away from $y$. Formally,
$$\mathrm{MSG}_{i \to j}(\ell_i, \tau) = \min_{z_{T_i} : z_i = \ell_i, \rho(z_{T_i}, y) \leq \tau} f(z_{T_i})$$
where $\ell_i \in \mathcal{L}$ and $\tau \in [0, \delta]$. This message cannot be computed until the messages from all children of $i$ have been computed. For ease of exposition, we add a pseudo vertex $s$ as the parent of the root, $r$. By definition, $\min_{\ell_r} \mathrm{MSG}_{r \to s}(\ell_r, \delta)$ is the potential of the desired labeling, $y^*$. Using the standard backtracking strategy of message passing, we can recover $y^*$. Please refer to [7] for details of the computation of each individual message. For convenience we call this procedure `Is-a-Mode`. This procedure and its variations will be used later.

### 3.2 Local and global modes

Given a graph $G$ and a collection of its subgraphs $\mathcal{B}$, we show that under certain conditions, there is a tight connection between the modes of these subgraphs and the modes of $G$. In particular, any consistent combinations of these local modes is a global mode, and vice versa.

Simply considering the modes of a subgraph $B$ is insufficient. A mode of $B$ with small potential may cause big penalty when it is extended to a labeling of the whole graph. Therefore, when defining a local mode, we select a boundary region of the subgraph and consider all possible label configurations of this boundary region. Formally, we divide the vertex set of $B$ into two disjoint subsets, the boundary $\partial B$ and the interior $\mathrm{int}(B)$, so that any path connecting an interior vertex $u \in \mathrm{int}(B)$ and an outside vertex $v \notin B$ has to pass through at least one boundary vertex $w \in \partial B$. See Figure 2(left) for examples of $B$. Similar to the definition of a global mode, we define a local mode as the partial labeling with the smallest potential in its $\delta$-neighborhood:

**Definition 2** (local modes)**.** *A partial labeling, $x_B$, is a local mode w.r.t. $\delta$-neighborhood if and only if there is no other partial labeling $y_B$ which (C1) has a smaller potential, $f(y_B) < f(x_B)$; (C2) is within $\delta$ distance from $x_B$, $\rho(y_B, x_B) \leq \delta$ and (C3) has the same boundary labeling, $y_{\partial B} = x_{\partial B}$.*

We denote by $\mathcal{M}_B^\delta$ the space of local modes of the subgraph $B$. Given a set of subgraphs $\mathcal{B}$ together with a interior-boundary decomposition for each subgraph, we have the following theorem.

**Theorem 3.1** (local-global). *Suppose any connected subgraph $G' \subseteq G$ of size $\delta$ is contained within $\mathrm{int}(B)$ of some $B \in \mathcal{B}$. A labeling $x$ of $G$ is a global mode if and only if for every $B \in \mathcal{B}$, the corresponding partial labeling $x_B$ is a local mode.*

*Proof.* The necessity is obvious since a global mode is a local mode within every subgraph. Note that necessity is not true any more if the restriction on $\partial B$ (C3 in Definition 2) is relaxed. Next we show the sufficiency by contradiction. Suppose a labeling $x$ is a local mode within every subgraph, but is not a global mode. By definition, there is $y \in \mathcal{N}_\delta(x)$ with smaller potential than $x$. We assume $y$ and $x$ disagree within a connected subgraph. If $y$ and $x$ disagree within multiple connected components, we can always find $y' \in \mathcal{N}_\delta(x)$ with smaller potential which disagree with $x$ within only one of these connected components. The subgraph on which $x$ and $y$ disagree must be contained by the interior of some $B \in \mathcal{B}$. Thus $x_B$ is not a local mode due to the existence of $y_B$. Contradiction. $\square$

We say partial labelings of two different subgraphs are *consistent* if they agree at all common vertices. Theorem 3.1 shows that there is a bijection between the set of global modes and the set of consistent combinations of local modes. This enables us to compute global modes by first compute local modes of each subgraph and then search through all their consistent combinations.

**Instantiating for a tree-structured graph.** For a tree-structured graph with $D$ nodes, let $\mathcal{B}$ be the set of $D$ geodesic balls, centered at the $D$ nodes. Each ball has radius $r = \lfloor \frac{\delta}{2} \rfloor + 1$. Formally, we have $B_i = \{j \mid \mathrm{dist}(i,j) \leq r\}$, $\partial B_i = \{j \mid \mathrm{dist}(i,j) = r\}$, and $\mathrm{int}(B_i) = \{j \mid \mathrm{dist}(i,j) < r\}$. Here $\mathrm{dist}(i,j)$ is the number of edges between the two nodes. See Figure 2(left) for examples. It is not hard to see that any size $\delta$ subtree is contained within a $\mathrm{int}(B_i)$ for some $i$. Therefore, the prerequisite of Theorem 3.1 is guaranteed.

We construct a junction tree to combine the set of all consistent local modes. It is constructed as follows: Each supernode of the junction tree corresponds to a geodesic ball. Two supernodes are neighbors if and only if their centers are neighbors in the original tree. See Figure 2(left). Let the label set of a supernode be its corresponding local modes, as defined in Definition 2. We construct a potential function of the junction tree so that a labeling of the junction tree has finite potential if and only if the corresponding local modes are consistent. Furthermore, whenever the potential of a junction tree labeling is finite, it is equal to the potential of the corresponding labeling in the original graph. This construction can be achieved using a standard junction tree construction algorithm, as long as the local mode set of each ball is given.

The $M$-modes problem is then reduced to computing the $M$ lowest potential labelings of the junction tree. This is the $M$-best labeling problem and can be solved efficiently using Nilsson's algorithm [18]. The algorithm of this section is summarized in the Procedure `Compute-M-Modes`.

---

**Procedure 1** `Compute-M-Modes`

---

**Input:** A tree $G$, a potential function $f$ and a scale $\delta$
**Output:** The $M$ modes of the lowest potential
 1: Construct geodesic balls $\mathcal{B} = \{B_r(c) \mid c \in \mathcal{V}\}$, where $r = \lfloor \frac{\delta}{2} \rfloor + 1$
 2: **for all** $B \in \mathcal{B}$ **do**
 3:    $\mathcal{M}_B^\delta$ = the set of local modes of $B$
 4: Construct a junction tree (Figure 2). The label set of each supernode is its local modes.
 5: Compute the $M$ lowest-potential labelings of the junction tree, using Nilsson's algorithm.

---

### 3.3 Computing local modes via enumeration

It remains to compute all local modes of each geodesic ball $B$. We give two different algorithms in Sec. 3.3 and 3.4. Both methods have two steps. First, compute a set of candidate partial labelings. Second, choose from these candidates the ones that satisfy Definition 2. In both methods, it is essential to ensure the candidate set contains all local modes.

**Computing a candidate set.** The first method enumerates through all possible labelings of the boundary. For each boundary labeling $x_{\partial B}$, we compute a corresponding subset of candidates. Each candidate is the partial labeling of the minimal potential with boundary labeling $x_{\partial B}$ and a fixed label $\ell$ of the center $c$. This subset has $L$ elements since $c$ has $L$ labels. Formally, the candidate subset for a fixed boundary labeling $x_{\partial B}$ is $\mathcal{C}_B(x_{\partial B}) = \{\mathrm{argmin}_{y_B} f_B(y_B) \mid y_{\partial B} = x_{\partial B}, y_c \in \mathcal{L}\}$. It can be computed using a standard message-passing algorithm over the tree, using $c$ as the root.

Denote by $\mathcal{X}_B$ and $\mathcal{X}_{\partial B}$ the space of all partial labelings of $B$ and $\partial B$ respectively. The candidate set we compute is the union of candidate subsets of all boundary labelings, i.e. $\mathcal{C}_B =$

$\bigcup_{x_{\partial B} \in \mathcal{X}_{\partial B}} \mathcal{C}_B(x_{\partial B})$. See Figure 2(right-bottom) for an example candidate set. We can show that the computed candidate set $\mathcal{C}_B$ contains all local modes of $B$.

**Theorem 3.2.** *Any local mode $y_B$ belongs to the candidate set $\mathcal{C}_B$.*

Before proving the theorem, we formalize an assumption of the geodesic balls.

**Assumption 1** (well-centered). *We assume that after removing the center from $\mathrm{int}(B)$, each connected component of the remaining graph has a size smaller than $\delta$.*

For example, in Figure 2(right-top), a geodesic ball of radius 3 has three connected components in $\mathrm{int}(B)\backslash\{c\}$, of size one, two and three, respectively. Since $r = \lfloor \frac{\delta}{2} \rfloor + 1$, $\delta$ is either four or five. The ball is well-centered. Since the interior of $B$ is essentially a ball of radius $r - 1 = \lfloor \frac{\delta}{2} \rfloor$, the assumption is unlikely to be violated, as we observed in practice. In the worst case when the assumption is violated, we can still solve the problem by adding additional centers in the middle of these connected components. Next we prove the theorem.

*Proof of Theorem 3.2.* We prove by contradiction. Suppose there is a local mode $y_B \notin \mathcal{X}_B(x_{\partial B})$ such that $y_{\partial B} = x_{\partial B}$. Let $\ell$ be the label of $y_B$ at the center $c$. Let $y'_B \in \mathcal{X}_B(x_{\partial B})$ be the candidate with the same label at the center. Furthermore, the two partial labelings agree at $\partial B$ and at $c$. Therefore the two labelings differ at a set of connected subgraphs. Each of the subgraphs has a size smaller than $\delta$, due to Assumption 1. Since $y'_B$ has a smaller potential than $y_B$ by definition, we can find a partial labeling $y''_B$ which only disagree with $y_B$ within one of these components. And $y''_B$ has a smaller potential than $y_B$. Therefore $y_B$ cannot be a local mode. Contradiction. $\square$

**Verifying each candidate.** Next, we show how to check whether a candidate is a local mode. For a given boundary labeling, $x_{\partial B}$, we denote by $\mathcal{X}_B(x_{\partial B})$ the space of all partial labelings with fixed boundary labeling $x_{\partial B}$. By definition, a candidate $y_B \in \mathcal{X}_B(x_{\partial B})$ is a local mode if and only if there is no other partial labeling in $\mathcal{X}_B(x_{\partial B})$ within $\delta$ from $y_B$ with a smaller potential. The verification of $y_B$ can be transformed into a global mode verification problem and solved by the algorithm in Sec. 3.1. We use the subgraph $B$ and its potential to construct a new graph. We need to ensure that only labelings with the fixed boundary labeling $x_{\partial B}$ are considered in this new graph. This can be done by enforcing each boundary node $i \in \partial B$ to have $x_i$ as the only feasible label.

### 3.4 Computing local modes using local modes of smaller scales

In Sec. 3.3, we computed the candidate set by enumerating all boundary labelings $x_{\partial B}$. In this subsection, we present an alternative method when the local modes of the scale $\delta - 1$ has been computed. We construct a new candidate set using local modes of scale $\delta - 1$. This candidate set is smaller that the candidate set from the previous subsection and thus leads to a more efficient algorithm. Since our algorithm computes modes from small scale to large scale. This algorithm can be used in all scales except for $\delta = 1$. The step of verifying whether each candidate is a local mode is the same as the previous subsection.

The following notations will prove convenient. Denote by $r$ and $r'$ the radii of balls for scales $\delta$ and $\delta - 1$ respectively (See Sec. 3.2 for the definition). Denote by $B_i$ and $B'_i$ the balls centered at node $i$ for scales $\delta$ and $\delta - 1$. Let $\mathcal{M}^\delta_{B_i}$ and $\mathcal{M}^{\delta-1}_{B'_i}$ be their sets of local modes at scales $\delta$ and $\delta - 1$ respectively. Our idea is to use $\mathcal{M}^{\delta-1}_{B'_i}$'s to compute a candidate set containing $\mathcal{M}^\delta_{B_i}$.

Consider two different cases, $\delta$ is odd and even. When $\delta$ is odd, $r = r'$ and $B_i = B'_i$. By definition, $\mathcal{M}^\delta_{B_i} \subseteq \mathcal{M}^{\delta-1}_{B_i} = \mathcal{M}^{\delta-1}_{B'_i}$. We can directly use the local modes of the previous scale as the candidate set for the current scale. When $\delta$ is even, $r = r' + 1$. The ball $B_i$ is the union of the $B'_j$'s for all $j$ adjacent to $i$, $B_i = \bigcup_{j \in N_i} B'_j$, where $N_i$ is the set of neighbors of $i$. We collect the set of all consistent combinations of $\mathcal{M}^{\delta-1}_{B'_j}$ for all $j \in N_i$ as the candidate set. This set is a superset of $\mathcal{M}^\delta_{B_i}$, because a local mode at scale $\delta$ has to be a local mode at scale $\delta - 1$.

**Dropping unused local modes.** In practice, we observe that a large amount of local modes do not contribute to any global mode. These unused local modes can be dropped when computing global modes and when computing local modes of larger scales. To check if a local mode of $B_i$ can be dropped, we compare it with all local modes of an adjacent ball $B_j$, $j \in N_i$. If it is not consistent with any local mode of $B_j$, we drop it. We go through all adjacent balls $B_j$ in order to drop as many local modes as possible.

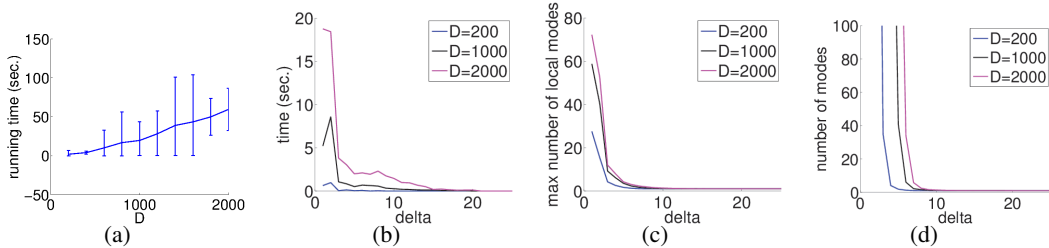

Figure 3: Scalability.

## 3.5 Complexity

There are three steps in our algorithm for each fixed $\delta$: computing, verifying candidates and computing the $M$ best labelings of the junction tree. Denote by $d$ the tree degree. Denote by $\lambda$ the maximum number of undropped local modes for any ball $B$ and scale $\delta$. When $\delta = 1$, we use the enumeration method. Since the ball radius is 1, the ball boundary size is $O(d)$. There are at most $L^d$ many candidates for each ball. When $\delta > 1$, we use local modes of the scale $\delta - 1$ to construct the candidate set. Since each ball of scale $\delta$ is the union of $O(d)$ many balls of scale $\delta - 1$, there are at most $\lambda^d$ many candidates per node. The verification takes $O(DdL\delta^2(L+\delta))$ time per candidate. (See [7] for complexity analysis of the verification algorithm.) Therefore overall the computation and verification of all local modes for all $D$ balls is $O(D^2 dL\delta^2 (L + \delta)(L^d + \lambda^d))$. The last step runs Nilsson's algorithm on a junction tree with label size $O(\lambda)$, and thus takes $O(D\lambda^2 + MD\lambda + MD\log(MD))$. Summing up these complexities gives the final complexity.

**Scalability.** Even though our algorithm is not polynomial to all relevant parameters, it is efficient in practice. The complexity is exponential to the tree degree ($d$). However, in practice, we can enforce an upperbound of the tree degree in the model estimation stage. This way we can assume $d$ to be constant. Another parameter in the complexity is $\lambda$, the maximal number of undropped local modes of a geodesic ball. When the scale $\delta$ is large, $\lambda$ could be exponential to the graph size. However, in practice, we observe that $\lambda$ decreases quickly as $\delta$ increases. Therefore, our algorithm can finish in a reasonable time. See Sec. 4 for more discussions.

## 4 Experiment

To validate our method, we first show the scalability and accuracy of our algorithm in synthetic data. Furthermore, we demonstrate using biological data how modes can be used as a novel analysis tool. Quantitative analysis of modes reveals new insight of the data. This finding is well supported by a visualization of the modes, which intuitively outlines the topographical map of the distribution. In all experiments, we choose $M$ to be 500. At bigger scales, there are often less than $M$ modes in total. As mentioned earlier, modes can also be applied to the problem of multiple predictions [7].

**Scalability.** We randomly generate tree-structured graphical model (tree size $D =200 \ldots 2000$, label size $L = 3$) and test the speed. For each tree size, we generates 100 random data. In Figure 3(a), we show the running time of our algorithm to compute modes of all scales. The running time is roughly linear to the graph size. In Figure 3(b) we show the average running time for each delta when the graph size is 200, 1000 and 2000. As we see most of the computation time is spent on computations with $\delta = 1$ and 2. Note only when $\delta = 1$, the enumeration method is used. When $\delta \geq 2$, we reuse local modes of previous $\delta$. The algorithm speed depends on the parameter $\lambda$, the maximum number of undropped local modes over all balls. In Figure 3(c), we show that $\lambda$ drops quickly as the scale increases. We believe this is critical to the overall efficiency of our method. In Figure 3(d), we show the average number of global modes at different scales.

**Accuracy.** We randomly generate tree-structured distributions ($D = 20$, $L = 2$). We select the trees with strong modes as ground-truth trees, i.e. those with at least two modes up to $\delta = 7$. See Figure 4(a) for the average number of modes at different scales over these selected tree models. Next we sample these trees and then use the samples to estimate a tree model to approximate this distribution. Finally we compute modes of the estimated tree and compare them to the modes of the ground-truth trees.

To evaluate the sensitivity of our method to noise, we randomly flip $0\%, 5\%, 10\%, 15\%$ and $20\%$ labels of these samples. We compare the number of predicted modes to the number of true modes for each scale. The error is normalized by the number of true modes. See Figure 4(b). With small noise, our prediction is accurate except for $\delta = 1$, when the number of true modes is very large. As the noise level increases, the error increases linearly. We do notice an increase of error at near $\delta = 7$. This is because at $\delta = 8$, many data become unimodal. Predicting two modes leads to $50\%$ error.

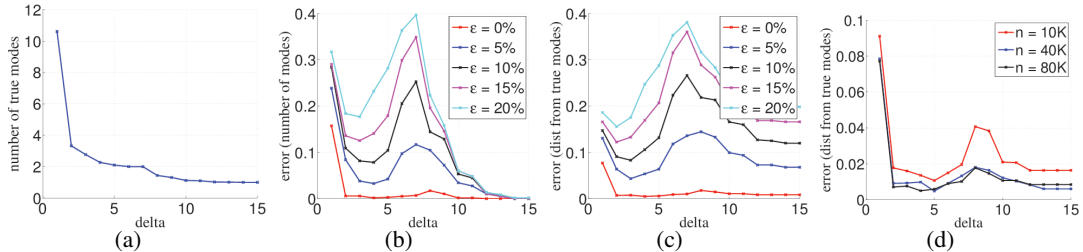

Figure 4: Accuracy. Denote by $\epsilon$ the noise level, $n$ the sample size.

We also measure the prediction accuracy using the Hausdorff distance between the predicted modes and the true modes. The Hausdorff distance between two finite points sets $X$ and $Y$ is defined as $\max\left(\max_{x \in X} \min_{y \in Y} \rho(x, y), \max_{y \in Y} \min_{x \in X} \rho(x, y)\right)$. The result is shown in Figure 4(c). We normalize the error using the tree size $D$. So the error is between zero and one. The error is again increasing linearly w.r.t. the noise level. An increase at $\delta = 7$ is due to the fact that many data change from multiple modes to one single mode. In Figure 4(d), we compare for a same noise level the error when we use different sample sizes. When the sample size is 10K, we have bigger error. When the sample size is 80K and 40K, the errors are similar and small.

**Biological data analysis.** We compute modes of the microarray data of Arabidopsis thaliana plant (108 samples, 39 dimensions) [24]. Each gene has three labels: "+", "0" and "-" respectively denote over-expression, normal-expression and under-expression of the genes. Based on the data sample we estimate the tree graph and compute the top modes with different radiuses $\delta$ using Hamming distance. We use multidimensional scaling to map these modes so that their pairwise Hamming distance is approximated by the $L_2$ distance in $\mathbb{R}^2$. The result is visualized in Fig. 5 with different scales. The size of the points is proportional to the log of its probability. Arrows in the figure show how each mode merges to survived modes at the larger scale. The graph intuitively shows that there are two major modes when viewed from a large scale and even shows how the modes evolve as we change the scale.

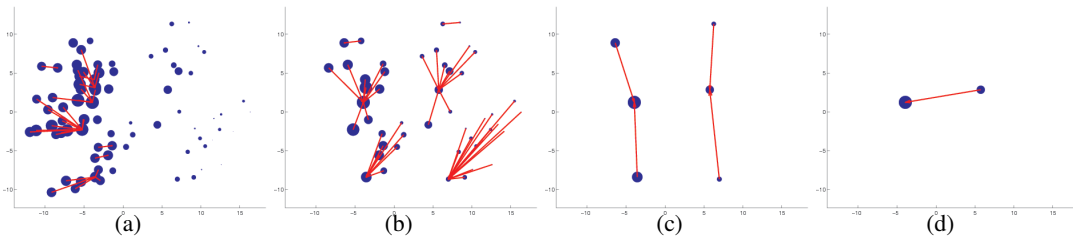

Figure 5: Microarray results. From left to right: scale 1 to 4.

## 5 Conclusion

This paper studies the $(\delta, \rho)$-mode estimation problem for tree graphical models. The significance of this work lies in several aspects: (1) we develop an efficient algorithm to illustrate the intrinsic connection between structured statistical modeling and mode characterization; (2) our notion of $(\delta, \rho)$-modes provides a new tool for visualizing the topographical information of complex discrete distributions. This work is the first step towards understanding the statistical and computational aspects of complex discrete distributions. For future investigations, we plan to relax the tree graphical model assumption to junction trees.

**Acknowledgments**

Chao Chen thanks Vladimir Kolmogorov and Christoph H. Lampert for helpful discussions. The research of Chao Chen and Dimitris N. Metaxas is partially supported by the grants NSF IIS 1451292 and NSF CNS 1229628. The research of Han Liu is partially supported by the grants NSF IIS1408910, NSF IIS1332109, NIH R01MH102339, NIH R01GM083084, and NIH R01HG06841.

## Footnotes

[1]For convenience, we drop unary potentials $f_i$ in this paper. Note that any potential function with unary potentials can be rewritten as a potential function without them.

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
