[Reviews · NeurIPS 2014]

Submitted by Assigned_Reviewer_17

The paper proposes an algorithm for estimating the (\delta,\rho) modes of a distribution. The algorithm approximates the distribution using a tree graphical model, constructed using samples from the original distribution, and finds the modes of the tree graphical model. An algorithm for finding the modes of a tree graphical models is proposed, based on constructing an appropriate junction tree to enforce constraints. The algorithm runs in polynomial time in most parameters but in exponential time to the degree of the tree. The method appears to be fast in experiments.

The paper is clearly written. The idea of using a tree distribution to approximate the original distribution is a sensible one, and the authors shows that an effective practical algorithm can be developed from that idea. The method appears to give a good tradeoff between quality and complexity and may potentially be quite useful in practice. The paper also provides analysis on the running time and theoretical analysis on performance when the distribution can be well approximated by tree distributions. Nice work overall.

Summary: The paper proposes a reasonable algorithm for (\delta,\rho) mode estimation that may turn out to be practically effective. Analysis of running time and error from sampling is provided with good experimental results are reported. Good paper overall.

Submitted by Assigned_Reviewer_24

Summary: The paper proposes an exact algorithm for the problem of finding the M best modes (local optima) for discrete tree-structured graphical models. The main idea is to first find the local M-modes on subgraphs that forms a covering of the original graph, and then combine the local modes together to form global modes.
Numerical experiments on both simulated and real datasets are presented. In particular, the biological data example forms an interesting illustration of the use of the local modes at different scales. Concentration results are also presented.

The paper is well written. The technique is solid. The algorithm and its extensions can be potentially useful tools for analyzing and visualizing high dimension discrete distributions.

Summary: The paper proposes an exact algorithm for finding M best local optima for tree structured discrete graphical models. Comprehensive experiments are presented to demonstrate its efficiency and practical applications.

Submitted by Assigned_Reviewer_42

Description:

The paper presents an efficient algorithm to compute modes of the
probability distribution over many discrete variables defined by a
tree-structured graphical model. More precisely, given a (eg Hamming)
metric and a constant delta, a labeling is a delta-mode if it has
greater probability than all other labelings within distance
delta. The proposed algorithm computes M best modes for each scale
delta. The complexity of the algorithm is exponential in theory but
reasonable empirically.

The algorithm uses a newly derived relation between local and global
modes. A local delta-mode is a labeling on a subgraph whch has higher
probability than all other labelings on the subgraph within distance
delta and with the same labeling on the boundary. Given a rich enough
collection of subgraphs of the original graph, a labeling is a global
delta-mode iff its restrictions to subgraphs are local
delta-modes. This implies that local delta-modes that are consistent
(agree on overlapping variables) define a global mode.

For a tree-structured model, the collection of subgraphs has a
polynomial size (they are delta-balls). The algorithm first finds all
local delta-modes for each subgraph. This is done by enumerating all
boundary labelings and then finding the maximum probability for each
such labeling (there is a non-trivial insight in this algorithm,
though). Then, the algorithm constructs a junction tree graphical
model whose nodes are the subgraphs, labels are the local modes, and
whose probability is the probability of the global mode if the local
modes are consistent and zero otherwise. Now, M best labelings
(computed by the existing algorithm [16]) in this junction tree model
corresponds to M best delta-modes in the original tree. Thus, the
problem of finding M best modes has been reduced to the known problem
of finding M best labelings.

When increasing delta, a mode can only disappear but never
appear. Modes for various delta thus form somethng like a scale space.
When we want to compute the modes for all scales delta, the modes for
some delta can be computed efficiently from already computed modes for
delta-1. Thus, the above costly enumeration must be done only for
delta=1.

Besides the algorithm, the paper presents theoretical guarantees that
delta-modes are stable under errors in estimating the tree from
samples, using the method from [1].

The algorithm is verified in several experiments. One uses synthetic
data, the other real biological data, and the third (in the
supplement) image segmentation from superpixels by CRF. The
experiments suggest that the number of local as well as global modes
is managable in practice. The runtime is tens of seconds for trees
with 2000 nodes. On image segmentation, the method almost always yields better results than the M-best diverse labeling method by [Batra et al, ECCV'12].

Comments:

This is very good paper with a lot of work behind it. It is a good
step towards practical algorithms for finding multiple modes of a
graphical model. I have no major objections. Minor objections are as
follows.

In Figure 3a, the uncertainty bars are wrong because runtime cannot be
negative.

To my understanding, for a fixed delta the algorithm finds not all
delta-modes but only M best delta-modes. But you never state what M was
in your experiments.

The text contains a number of very minor language mistakes, such as
"it is a local optima" (l.17), "inversely logarithm proportional"
(l.107).

On l.247: Why don't you write y_c\in L istead of y_c=\ell, \ell\in L.
Summary: A very good paper on a useful topic.
Author Feedback
Author rebuttal: We thank all reviewers for helpful suggestions. Here we address a few minor comments.

1. Minor mistakes in Fig. 3a, l.17, l.107, l.247.
Answer: Thanks for pointing them out. We will fix them accordingly.

2. What is $M$ in the experiments?
Answer: For all experiments in Sec. 5, we choose $M$ to be 500. In practice, a large $M$ does not hurt the running time much. One reason is that when $\delta$ is large, the total number of modes is typically much smaller than 500.

In the segmentation task (Supplemental Material), we set $M$ to 30. The main reason is that the competing baseline (NMS) cannot finish when $M$ is large.

We will clarify this in the final version.